# NuA4 and SAGA acetyltransferase complexes cooperate for repair of DNA breaks by homologous recombination

**Xue Cheng , Valérie Côté, Jacques Côté ***

St-Patrick Research Group in Basic Oncology, Laval University Cancer Research Center, Oncology Division of CHU de Québec-Université Laval Research Center, Quebec City, Canada

* jacques.cote@crhdq.ulaval.ca

## Abstract

Chromatin modifying complexes play important yet not fully defined roles in DNA repair processes. The essential NuA4 histone acetyltransferase (HAT) complex is recruited to double-strand break (DSB) sites and spreads along with DNA end resection. As predicted, NuA4 acetylates surrounding nucleosomes upon DSB induction and defects in its activity correlate with altered DNA end resection and Rad51 recombinase recruitment. Importantly, we show that NuA4 is also recruited to the donor sequence during recombination along with increased H4 acetylation, indicating a direct role during strand invasion/D-loop formation after resection. We found that NuA4 cooperates locally with another HAT, the SAGA complex, during DSB repair as their combined action is essential for DNA end resection to occur. This cooperation of NuA4 and SAGA is required for recruitment of ATP-dependent chromatin remodelers, targeted acetylation of repair factors and homologous recombination. Our work reveals a multifaceted and conserved cooperation mechanism between acetyltransferase complexes to allow repair of DNA breaks by homologous recombination.

## Author summary

DNA double-strand breaks (DSBs) are among the most dangerous types of DNA lesions as they can produce genomic instability that leads to cancer and genetic diseases. It is therefore crucial to understand the precise molecular mechanisms used by cells to detect and repair this type of damages. Homologous recombination using sister chromatid as template is the most accurate pathway to repair these breaks but has to occur within the context of the DNA compacted structure in chromosomes. Here, we show that two enzymes, NuA4 and SAGA, that acetylate the structural components of chromosomes in the vicinity of the DNA breaks are together essential for recombination-mediated repair to occur. We found that they are recruited at an early step after damage detection and their action allows subsequent remodeling of local structural organisation by other enzymes, providing DNA access to the recombination machinery. These results highlight the cooperation of enzymes for a same goal, providing robustness in the repair process as only the loss of both leads to major defects.

**Data Availability Statement:** All relevant data are within the manuscript and its Supporting Information files.

**Funding:** This work was supported by a grant from the Canadian Institutes of Health Research (cihr-

irsc.gc.ca)(FDN-143314) to J.C. X.C. was
supported by doctoral studentships from Fonds de
Recherche du Québec-Santé (www.frqs.gouv.qc.
ca), Fondation du CHU de Québec-Université Laval
(fondationduchudequebec.org) and Centre de
Recherche sur le Cancer de l'Université Laval (crc.
ulaval.ca). J.C. holds the Canada Research Chair in
Chromatin Biology and Molecular Epigenetics
(www.chairs-chaires.gc.ca). The funders had no
role in study design, data collection and analysis,
decision to publish, or preparation of the
manuscript.

**Competing interests:** The authors have declared
that no competing interests exist.

## Introduction

DNA double-strand breaks (DSBs) are one of the most detrimental types of DNA lesions that
need to be handled properly to preserve genome integrity [1]. DSBs can be repaired by two
major pathways, non-homologous end-joining (NHEJ) and homologous recombination (HR),
while other types of repair also play important roles [2]. Repair by NHEJ tends to be error-
prone due to its template-independent nature whereas repair by HR is mostly error-free [2].
Notably, DSB repair takes place in the context of chromatin. Consistent with this, several stud-
ies have gradually delineated the functional interplay between chromatin modifiers/remode-
lers and critical processes during DNA repair [3,4].

The yeast 13-subunit NuA4 histone acetyltransferase (HAT) complex has been implicated
in the repair of DNA breaks [5–9]. Highly conserved during evolution, the main activity of
NuA4 is to acetylate nucleosomal histone H4 and H2A(Z/X) through its catalytic subunit
KAT5 (Esa1/Tip60), the sole HAT protein essential for cell viability in *Saccharomyces cerevi-
siae* [10,11]. Initial evidence for NuA4 involvement in repair came from yeast experiments
showing that cells with mutations of NuA4 subunits or its lysine targets on the H4 tail are sen-
sitive to DNA-damaging agents, suggesting that NuA4 and its activity are required for efficient
repair [6,8,12,13]. Subsequent analysis of NuA4 functions in DSB repair found that it is physi-
cally recruited to an endonuclease-induced DSB [6]. This recruitment is performed through a
direct interaction with the Mre11-Rad50-Xrs2 (MRX) complex and leads to spreading of
NuA4 along with DNA end resection [5,14]. Local acetylation of chromatin has been argued to
help recruitment of ATP-dependent remodelers near the break site [6,15]. NuA4 has also been
implicated in DNA damage tolerance, post-replicative and nucleotide excision repairs [16–
18], the latter also involving Gcn5/SAGA, another major HAT activity [19]. In human cells,
NuA4 (a.k.a. the TIP60 complex) regulates DNA damage signalling and counteracts 53BP1 to
favour repair by HR instead of NHEJ [20–22]. However, the functional consequences of NuA4
recruitment near DSBs in the repair process per se remain to be clearly defined.

In this study, we aimed to define NuA4 function and requirement during the repair of
DNA DSB. Using a conditional mutant and a rapid depletion approach in the context of inte-
grated DSB-induction system in budding yeast, we found that NuA4 participates in multiple
steps during HR but do so through an essential collaboration with another HAT complex,
SAGA, regulating chromatin dynamics, DNA end resection and non-histone substrates.

## Results

### NuA4-dependent acetylation of chromatin around a DNA break affects end resection

In order to study NuA4 function in DNA DSB repair, we first used a thermosensitive (ts)
mutant of Esa1 (*esa1-L254P*) [11] integrated in a well-established system that uses the HO
endonuclease under the control of the *GAL* promoter to produce a single inducible DSB at the
*MAT* locus in the yeast genome (p*GAL-HO*, *hmlΔ/hmrΔ*) [23]. Although the non-permissive
temperature compromised HO cleavage efficiency (**S1A Fig**) potentially due to NuA4 involve-
ment in transcription [10], semi-permissive temperature allows us to partially cripple NuA4
activity while still effectively generating DSBs *in vivo* (**S1B and S1C Fig**). Accordingly, the
increase of local H4 acetylation detected upon DSB formation measured by ChIP-qPCR at dif-
ferent distances from the break (normalised for nucleosome occupancy and compared to no
DSB conditions) is lost at semi-permissive temperature (**Figs 1A, 1B and S1D**). In parallel, we
also integrated the anchor-away system [24] into the aforementioned inducible DSB back-
ground, aiming to rapidly deplete proteins from the nucleus upon rapamycin treatment while

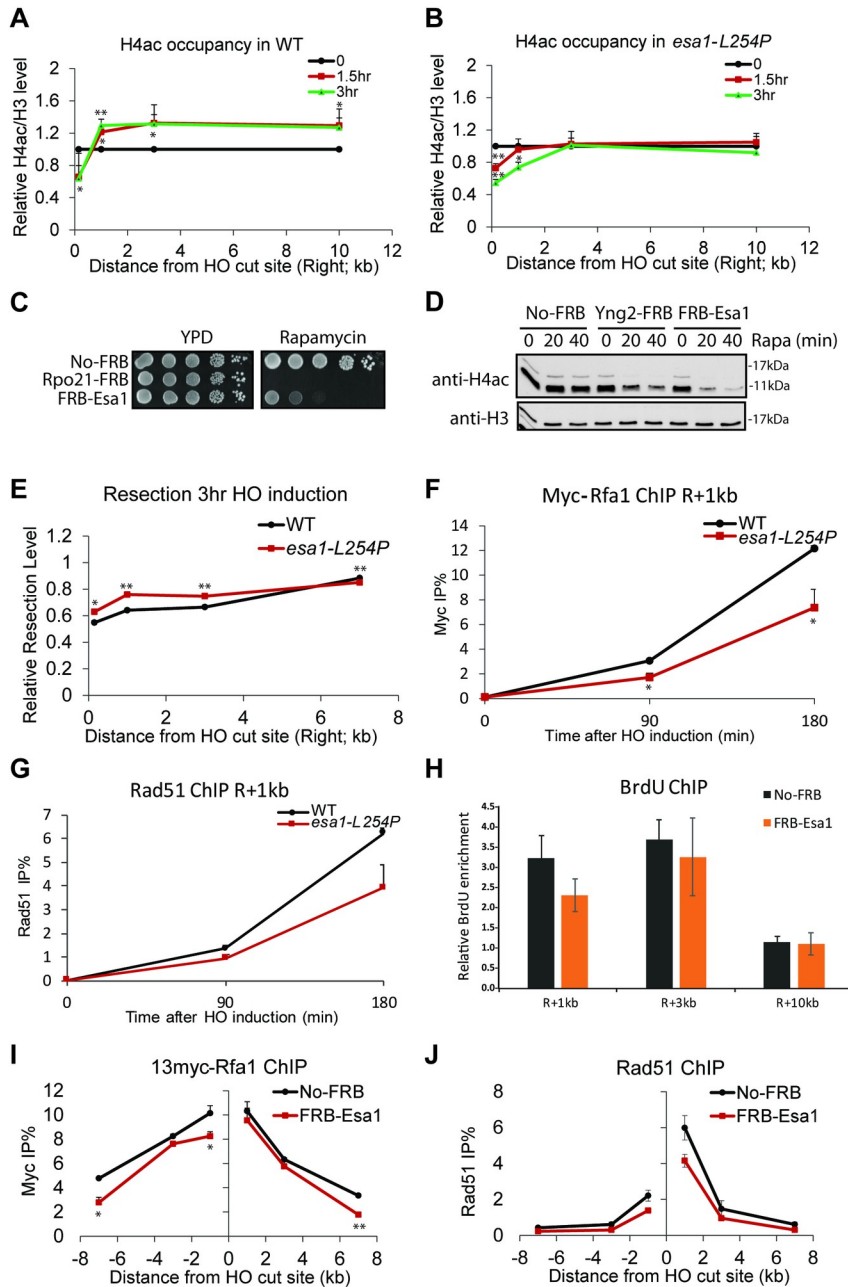

**Fig 1. NuA4 is required for H4 acetylation upon DSB induction and affects resection. A-B)** ChIP-qPCR of H4ac signal near the DSB site (H4ac/total H3 ratio of %IP/input) in WT and *esa1 ts* strains during 0, 90 and 180min of galactose induction of HO nuclease. Values at the different locations near the DSB at time 0 was normalized to 1, showing the increase of H4 acetylation upon HO cut in WT conditions but not in mutant at 32˚C (error bars are range of biological replicates). **C)** 10-fold serial dilutions of indicated FRB-tagged strains were spotted on YPD or YPD containing 1μg/ml rapamycin and grown at 30˚C, showing loss of viability by depletion of Esa1 and RNA PolII largest subunit for comparison (Rpo21). **D)** WB of WCE from indicated strains treated with 1μg/ml rapamycin for 0, 20 or 40min, showing loss of H4 acetylation by depletion of Esa1 and Yng2, another subunit of NuA4 (not used further as it affects Gal induction). **E)** Resection assay of WT and *esa1-L254P* cells after 3hr of galactose induction at 32˚C. DNA content around the break was measured by qPCR with genomic DNA and normalized on uncut locus (interV), showing higher level of DNA remaining near the DSB in mutant cells. **F-G)** ChIP-qPCR assay of 13myc-Rfa1 (F) and Rad51 (G) next to the DSB site in WT and *esa1-L254P* after 0, 90 and 180min of HO induction at 32˚C, showing less accumulation in mutant conditions. **H)** ChIP-qPCR assay of BrdU in No-FRB and FRB-Esa1 strains after 3hr of galactose induction in presence of rapamycin, showing slightly less exposed ssDNA near the break after Esa1 depletion. BrdU enrichment was normalized on control locus (interV). **I-J)** ChIP-qPCR assay of 13myc-Rfa1 (I) and Rad51 (J) in

No-FRB and FRB-Esa1 cells after 3hr galactose induction in presence of rapamycin, showing less accumulation in Esa1-depleted conditions. Error bars in ChIP-qPCR represent range from biological replicates. (Statistical analyses were performed by one-tailed t-test: * p<0.05, ** p<0.01, *** p<0.001).

importantly preserving robust DSB formation. Unlike other depletion approaches that involves degradation of the target protein, this system uses a rapamycin-induced physical interaction with a ribosomal protein to rapidly deplete a protein from the nucleus by forcing its translocation to the cytoplasm [24]. Since C-terminal tagging of Esa1 with FRB leads to a decrease in global acetylation even without rapamycin treatment (**S1E Fig**, potentially due to the importance of an intact C-terminus), we created N-terminal FRB-tagging of endogenous Esa1, keeping it under control of its own promoter. Upon rapamycin treatment, FRB-Esa1 cells show dramatic defect in growth (**Fig 1C**) and a rapid decrease in global H4 acetylation level (**Fig 1D**), demonstrating that anchor-away of FRB-Esa1 is potent and efficient (i.e. validating Esa1 depletion from the nucleus). Notably, the cutting efficiency of the HO endonuclease under these Esa1 anchor-away conditions is comparable to the respective WT (No-FRB) while H4 acetylation near the break is crippled (**S1F–S1I Fig**). Interestingly, both systems indicate that the domain of NuA4-dependent increased H4 acetylation upon DSB formation is larger than the extent of detected NuA4 binding/spreading in these conditions (3-5kb, see [5]), suggesting longer-range action in the tri-dimensional space, similar to what has been reported for chromatin remodelers [25].

Since histone acetylation can directly attenuate DNA-histone interactions and can be recognized by bromodomain-containing ATP-dependent chromatin remodelers [10,15], its NuA4-dependent increase near the DSB is expected to affect nucleosome stability [9](**S1J Fig**) and downstream events like HR-linked DNA end resection process. Nucleosomes represent physical barriers to resection, thus defects in nucleosome displacement generally impact repair of the break [3,26–30]. Accordingly, in the *ts* mutant, we detect a small but significant decrease of DNA end resection near the break site which is shown by the mutant having more DNA left near the break (red line) compared to WT conditions (black line) (resection is measured by a lower qPCR signal at different locations near the DSB due to the loss of one DNA strand) (Fig 1E). The mutant conditions also show decreased single-strand DNA (ssDNA) binding RPA (Rfa1 subunit) and Rad51 recombinase recruitment measured by ChIP-qPCR (**Fig 1F and 1G**). These results are also confirmed using the anchor-away system with ssDNA detection using BrdU-labeling followed by anti-BrdU ChIP-qPCR in non-denaturing conditions as well as RPA/Rad51 ChIP-qPCR (**Figs 1H–1J and S1K and S1L**).

A functional link between NuA4 and resection is further supported by strong genetic interactions detected between a viable anchor-away NuA4 mutant (Eaf1-FRB) and key resection factors in the presence of DNA damage, namely early acting/MRX-linked endonuclease Sae2, as well as longer and parallel acting exonuclease Exo1 and helicase Sgs1 [31](**Fig 2A–2D**). Accordingly, NuA4 does not affect DSB repair by resection-independent end-joining (**S2A Fig**). On the other hand, there is no increased DNA damage sensitivity with deletion of the Fun30 ATP-dependent chromatin remodeler that has been linked to DNA end resection [28], suggesting an epistatic relationship (**Figs 2E and S2B**). If DNA damage sensitivities observed in NuA4 mutant cells are solely due to the impaired resection, one would expect boosting resection activity would rescue NuA4 mutant sensitivity to DNA damage. Overexpression of long-range resection factor Exo1 has been shown to rescue the *FUN30* deletion [28]. However, we did not observe suppression of the DNA damage sensitivity of *esa1* mutant cells when overexpressing Exo1 (**Fig 2F**). This suggests that NuA4 role in the repair of DNA breaks is not limited to resection. While decreased recruitment of the Rad51 recombinase in Esa1-depleted

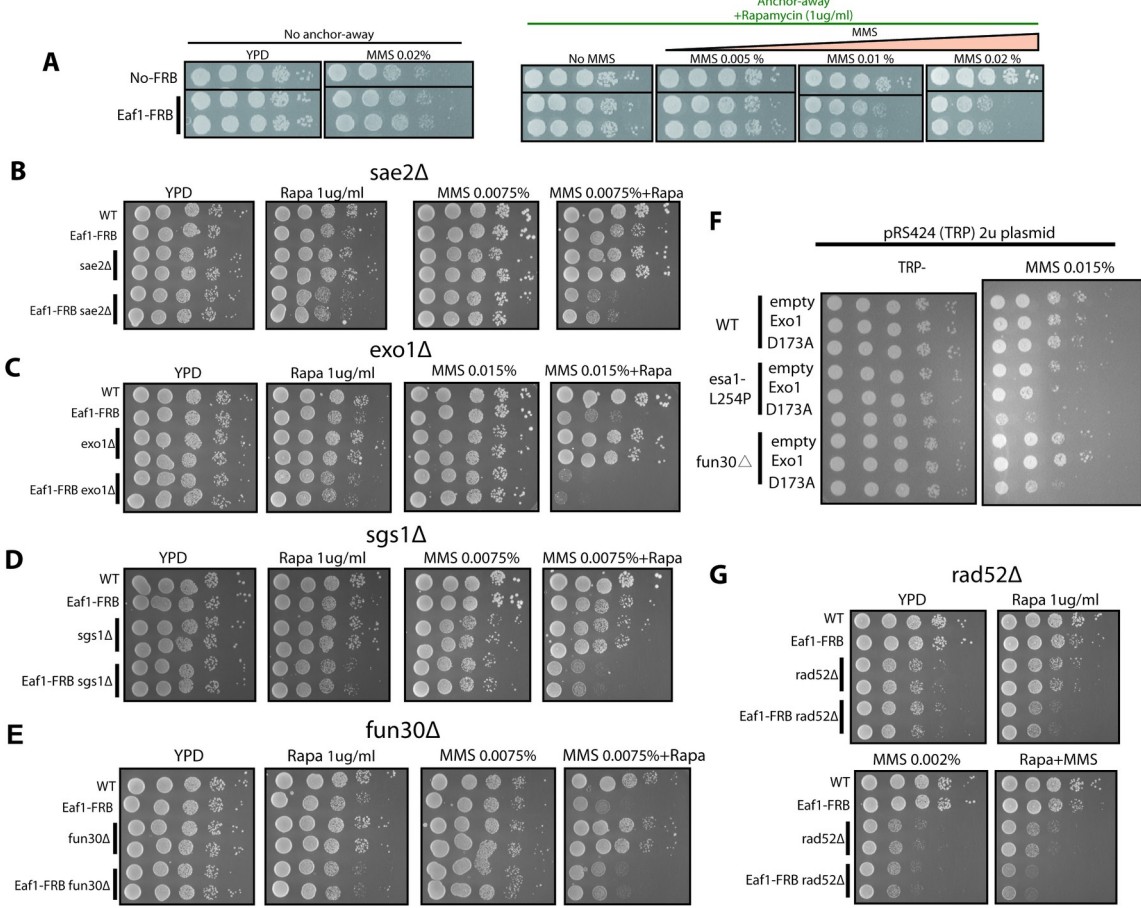

**Fig 2. NuA4 shows genetic interaction with resection and recombination factors. A-E, G)** 10-fold serial dilutions of indicated strains were spotted on solid medium without or with 1μg/ml rapamycin supplemented without or with indicated amounts of MMS and grown at 30°C. Another subunit of NuA4 was used with the FRB tag, scaffold subunit Eaf1[12], as its depletion does not affect cell viability on rich media (A). Genetic interaction (sickness/growth defect in presence of DNA damage induced by MMS) is seen with genes for resection factors Sae2 (B), Exo1 (C), Sgs1 (D) but not Fun30 (E), and with recombination factor Rad52 (G). **F)** DNA damage sensitivity of *esa1 ts* cells cannot be overcome by over-expression of potent resection factor Exo1. Spot assays as above with the indicated strains. The *fun30* deleted strain is used as control as it was shown to be rescued by Exo1 expression but not its catalytic mutant form (D173A) [28].

cells (**Fig 1G and 1J**) is most likely linked to reduced resection/RPA loading, genetic interaction between NuA4 and recombination factor Rad52 (**Fig 2G**) suggests that NuA4 may also play a role in the recombination step after resection.

## NuA4 plays a role in strand invasion at the homologous donor sequence

To progress through HR, the Rad51 recombinase replaces RPA on the resected ssDNA and this Rad51 filament performs homology search, ultimately invading the homologous donor sequence to form a structure termed D-loop. Importantly, this donor sequence is also covered with nucleosomes that require disruption. To investigate whether NuA4 can directly participate in the later steps of HR such as recombination/D-loop formation, we again took advantage of the inducible DSB system in this case carrying an intact donor sequence (*HML/HMR*) in the genome (**Fig 3A**). After break induction followed by removal of the HO endonuclease, we observed by ChIP the presence of both Rad51 and NuA4 at the invading strand, as

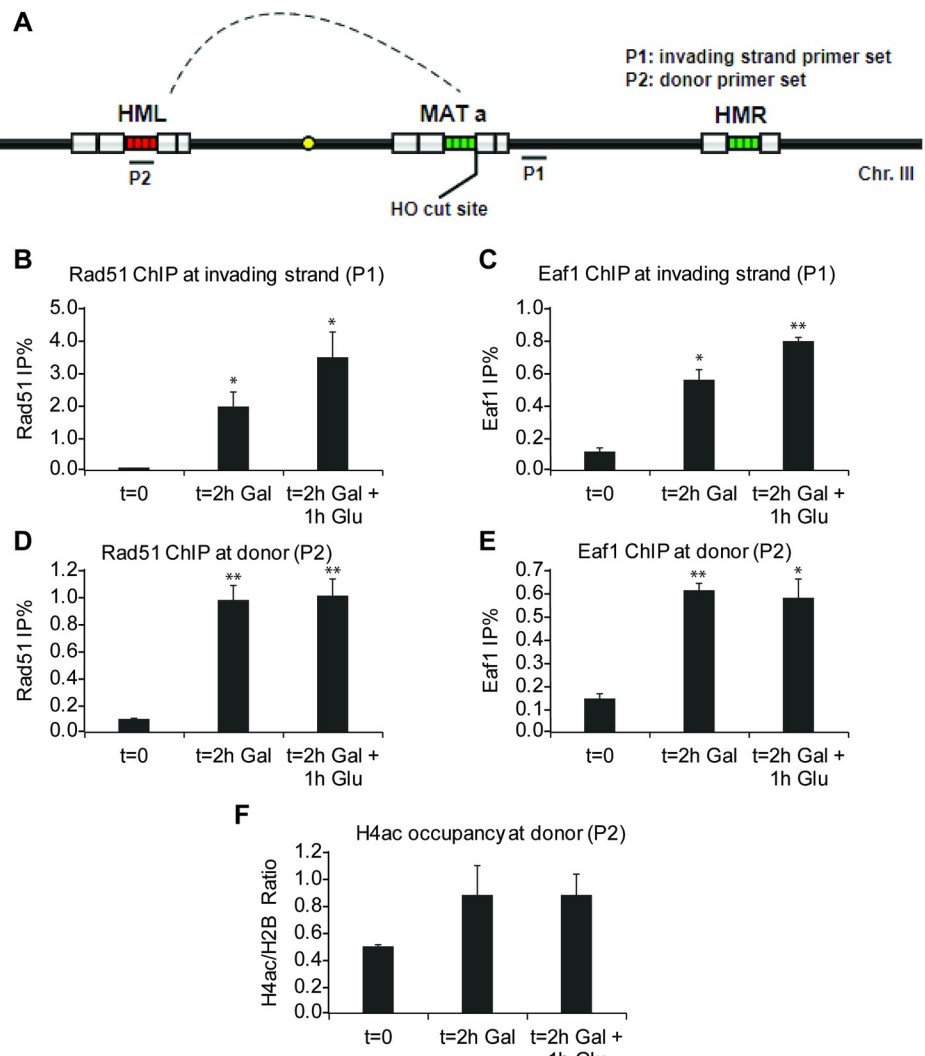

**Fig 3. NuA4 functions at the donor homologous sequence during HR, implicating chromatin acetylation during D-loop formation. A)** Schematic representation of HR donor strain genome arrangement and primer positions for ChIP-qPCR. **B-E)** ChIP-qPCR assay (% of Ip/input) of Rad51 (B, D) and Eaf1 (C, E) in HR donor strain after 0, 2hr galactose induction or 2hr galactose induction followed 1hr glucose (2%) treatment to block HO nuclease expression. Rad51 and NuA4 are clearly recruited at both the invading strand from the HO cut and and at the donor sequence during HR. Error bars represent range from biological duplicates. **F)** ChIP-qPCR of NuA4-dependent H4 acetylation at the donor sequence (H4ac/total H2B ratio of %IP/input), showing a clear increase during HR. Error bars represent standard error from biological triplicates. (Statistical analyses were performed by one-tailed t-test: * $p<0.05$, ** $p<0.01$, *** $p<0.001$; and multiple comparisons significance was also confirmed by one-way ANOVA followed by Tukey test).

previously reported (**Figs 3B** and **2C**) [5,32]. Importantly, the association of Rad51 was also detected using primers specific for the donor sequence, confirming the formation of D-loop (**Fig 3D**). Interestingly, the presence of NuA4 was also found at the donor locus (**Fig 3E**) (using an antibody for Eaf1, the only subunit unique to NuA4 [12]). Importantly, a concomitant increase of H4 acetylation is also detected at the donor sequence, albeit just short of statistical significance (p = 0.06) (**Fig 3F**). These results provide the first evidence that physically links NuA4 and its activity to formation of the D-loop, suggesting a role in modulating chromatin structure at the donor sequence to allow recombination.

To measure the importance of NuA4 in the recombination process *in vivo*, we used a yeast HR reporter assay in which an uncuttable *MAT* donor sequence (*MAT*-inc) is integrated in the genome of the inducible HO-DSB system. In this system, cells with DSBs repaired by HR will become resistant to HO cleavage, thus growing on the solid medium containing galactose [33]. Surprisingly, using both the *esa1-L254P* mutant at semi-permissive temperature and the viable anchor-away NuA4 mutant (Eaf1-FRB), we did not observe a significant defect in homologous recombination using this assay (**S2C and S2D Fig**). While this was also reported for another NuA4 mutant [13], one explanation may be that NuA4 itself is not essential for the repair of endonuclease-induced "clean-end" DSBs, similar to what has been reported for Sae2 (CtIP) and Mre11 nuclease [34]. Another possibility is the relatively small effect on end resection seen in NuA4 mutants not being sufficient to affect HR in this assay over the long time of growth on plates. It could also be due to functional redundancy or cooperativity with another chromatin modifier (see below).

## Combined recruitment of NuA4 and SAGA acetyltransferases is essential for DNA end resection

In a previous ChIP experiment coupled to mass spectrometry to identify proteins linked to DNA end resection (RPA) after HO-DSB formation, we could detect NuA4 components along with repair proteins [5]. Interestingly, beside ATP-dependent chromatin remodelers, a subunit of the SAGA histone acetyltransferase complex was also detected (**Fig 4A**). In transcription regulation, NuA4 has been shown to collaborate with SAGA, which targets H3 and H2B tails in nucleosomes through its catalytic subunit Gcn5 [35–37]. Both NuA4 and SAGA have also been shown to acetylate non-histone substrates, some linked to the DNA repair process [5,7,38–41]. NuA4 and SAGA share the large ATM/ATR-related Tra1 subunit which is implicated in their recruitment to gene promoters by activators [42]. Since NuA4 is recruited by the MRX complex potentially through a direct interaction with Tra1 [5], we asked if the SAGA complex is also recruited to DNA breaks and collaborates with NuA4, as in gene regulation. Indeed, using SAGA scaffold subunit Spt7 [43], we observe clear recruitment of SAGA by ChIP after DSB induction, covering 3-5kb next to the break (**Fig 4B**). Importantly, this recruitment of SAGA is lost in *xrs2Δ* mutant cells, suggesting that it also involves MRX. To test whether this similar mode of recruitment leads to a functional collaboration between NuA4 and SAGA during DSB repair by homologous recombination, we examined resection levels in Esa1 and/or Gcn5 anchor-away conditions by measuring the exposure of ssDNA using BrdU and RPA binding by ChIP-qPCR (**Figs 4C, 4D** and **S3A–S3D** for controls). Although anchor-away of a single HAT shows little difference compared to No-FRB cells, a complete loss in ssDNA exposure is observed when both HATs are depleted (**Fig 4C**), correlated by a similar complete loss of RPA binding (**Fig 4D**), supporting an essential combined role of NuA4 and SAGA during DNA end resection. As the ssDNA generated by resection can activate DNA damage checkpoint through the Mec1-dependent pathway [44], we reasoned that the crippled resection in double anchor-away conditions should also impact checkpoint activation. Consistent with our hypothesis, checkpoint activation is highly defective in double anchor-away conditions as shown by measuring Rad53 phosphorylation (**Fig 4E**). It has been argued that NuA4 and SAGA acetyltransferase activities can independently affect the recruitment of the SWI/SNF chromatin remodeler near DSBs through bromodomain-acetylated histones interactions [15]. Since the loss of acetylation correlates with increased nucleosome occupancy near the DSB in our Esa1/Gcn5-double depleted background (**S3E and S3F Fig**), we investigated defects in the recruitment of chromatin remodelers. Recruitment of the SWI/SNF complex is indeed greatly affected by the combined loss of NuA4 and SAGA (**Fig 4F and 4G**), but we also

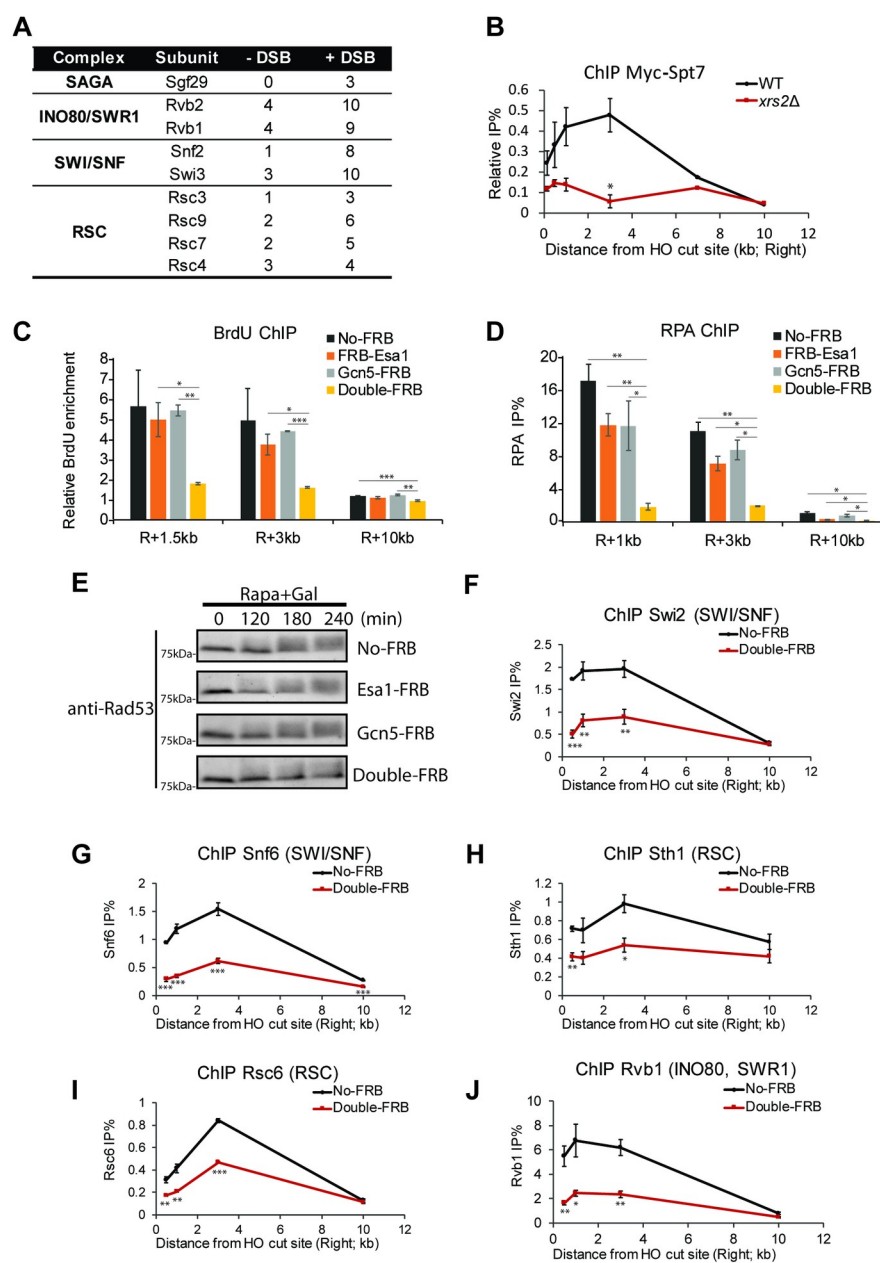

**Fig 4. Essential collaboration of NuA4 and SAGA HAT complexes at DSBs to allow DNA end resection and recruitment of ATP-dependent chromatin remodelers. A)** List of protein components of chromatin remodeling/modifying complexes and number of total peptides detected in ChIP of Flag-Rfa1 followed by MS analysis from cells after 0 or 90 min of HO induction [5], showing increased association of remodelers but also a component of the SAGA HAT complex. **B)** ChIP-qPCR assay of Myc-Spt7 (SAGA scaffold subunit) in WT and *xrs2Δ* cells after 3hr of HO induction with galactose. % of IP/input is shown at different locations near the DSB after subtraction of the signal in untagged strain. **C)** ChIP-qPCR assay of BrdU in indicated FRB-tagged strains after 3hr galactose induction in the presence of rapamycin to measure ssDNA production (as in Fig 1H). BrdU enrichment was normalized on control locus (inter V). **D)** ChIP-qPCR as in (B) but measuring resection through anti-RPA, showing again full loss of resection when Esa1 and Gcn5 are both depleted. **E)** DNA damage checkpoint activation measured by Western blotting of Rad53 (Santa cruz, sc-6749) from whole cell extracts after 0, 2, 3, 4hr of galactose induction in presence of rapamycin, showing slower migrating signal upon activation by DNA damage, which is lost after depletion of both Esa1 and Gcn5. **F-J)** Recruitment of ATP-dependent chromatin remodeling complexes at locations near the HO DSB as measured by ChIP-qPCR with specific antibodies after 3hr of galactose induction in presence of rapamycin. The Esa1/Gcn5 double depletion shows clear defects in recruitment of these factors. Error bars represent range from biological duplicates. (Statistical analyses were performed by one-tailed t-test: * p<0.05, ** p<0.01, *** p<0.001; and multiple comparisons significance for C and D was also confirmed by one-way ANOVA followed by Tukey test).

see defects in the recruitment of the essential remodeling complex RSC which contains multiple bromodomains (**Fig 4H and 4I**), as well as the Rvb-containing remodeling complexes (Ino80/Swr1)(**Fig 4J**).

## NuA4 cooperates with SAGA to allow HR-mediated repair of DSBs in eukaryotes

Using qPCR to measure repair by HR in the inducible DSB system coupled to an uncuttable donor sequence (as above), a strong defect in HR efficiency is also observed in Esa1/Gcn5 double anchor-away conditions (**Fig 5A and 5B**). Depletion of the potential SAGA/NuA4 common recruiting interface Tra1 by anchor-away also shows reduction in RPA and Rad51 association on each side of the DSB, albeit not as much as the double Esa1/Gcn5 depletion, suggesting other means of recruitment (**S4A–S4D Fig**). Altogether, these results indicate that NuA4 collaborates with SAGA to promote repair of DNA breaks by HR, at least in part through combined acetylation of chromatin (histone H4/H2A/H2B/H3 tails) that promotes nucleosome removal/disruption by chromatin remodelers.

We next asked whether NuA4 and SAGA could also collaborate in targeting non-histone substrates implicated in DNA repair, such as key resection/ssDNA-binding factor RPA, which has been shown to be a substrate of NuA4 in yeast DSB repair [5] and Gcn5 in mammalian nucleotide excision repair [39,41]. Acetylation of yeast RPA by NuA4 on multiple lysine residues has been shown to affect its binding to single-strand DNA produced during resection [5]. Using Acetyl-lysine immunoprecipitation (IP) followed by RPA western blot (WB), we observe decreased RPA signal in Esa1 or Gcn5 anchor-away conditions (**Fig 5C**). Importantly, RPA signal is further strongly decreased in the double anchor-away condition, indicating that RPA acetylation depends on both NuA4 and SAGA *in vivo*.

While mammalian NuA4, the TIP60 complex, has been implicated in break repair by HR for many years [20–22], the role of mammalian SAGA came to light more recently [45]. The well-conserved nature of both complexes led us to test whether they also function together to promote HR-mediated break repair in human cells. We used the DR-GFP HR reporter assay in U2OS cells by measuring GFP-positive cells after I-SceI induction of a DSB (as in [20]). While single siRNA-mediated knock-down (KD) of either Tip60 or Gcn5 shows mild HR defect, double KD led to a strong defect in HR efficiency (**Fig 5D**). KD of the Gcn5 paralog PCAF did not show this effect but a significant alteration of the cell cycle profile makes it difficult to interpret (**S5A and S5B Fig**). Overall, our results support an essential conserved cooperation between two major HAT complexes, NuA4 and SAGA, to promote repair of DNA breaks by homologous recombination, through acetylation of both chromatin and non-histone substrates.

## Discussion

In this study, we discovered that the NuA4 complex is involved not only in the resection but also at the recombination step of repair by HR. The role of NuA4-dependent acetylation of histone H4 and H2A in a large domain surrounding the DNA break was expected to be linked to nucleosome destabilization to allow end resection but it is now also linked to strand invasion/D-loop formation at the donor sequence (**Fig 3**). NuA4 mutants have been shown to affect recruitment of ATP-dependent chromatin remodelers INO80/SWR1 and bromodomain-containing SWI/SNF to the HO-induced break [6,14], as well as the RSC complex to stabilize GAG-repeats during replication [17]. The Gcn5 HAT has also been shown to affect SWI/SNF recruitment at the break [15] and, recently, to collaborate with MRX at stalled replication forks, playing a role in resection of nascent DNA [46]. We now show that the Gcn5-containing

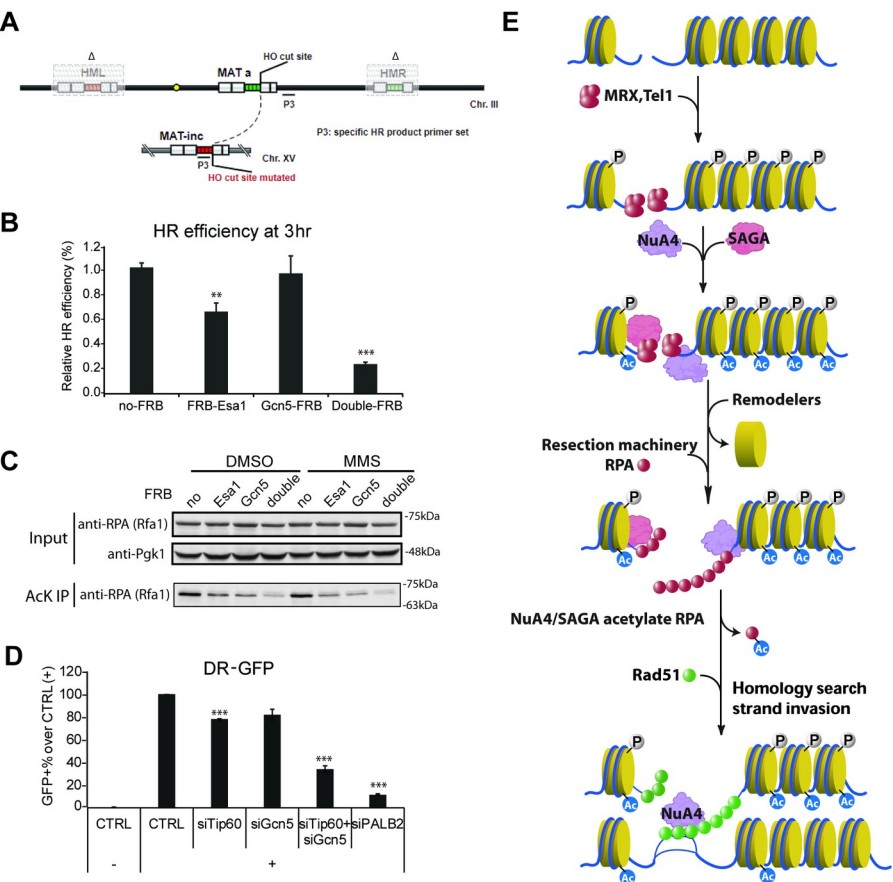

**Fig 5. NuA4 and SAGA also cooperate for acetylation of RPA and are together required of DSB repair by HR from yeast to humans. A)** Schematic representation of uncuttable-donor (*MAT*-inc) strain genome arrangement and primer positions to detect repair by HR. **B)** HR assay by qPCR on genomic DNA with primers shown (A) after 3hr of HO induction in presence of rapamycin and normalization on control locus (InterV). Double depletion of Esa1 and Gcn5 shows strong defect of repair by HR. Error bars represent standard error from three biological replicates. **C)** Measurement of *in vivo* RPA acetylation by Western blotting after acetyl-lysine IP from whole cell extracts corresponding to the indicated strains, incubated with rapamycin plus DMSO or MMS (0.05%) for 3hr. Inputs signals before Kac IP are shown for control as well as Pgk1 for loading and relative molecular weight marker positions. Both Esa1 and Gcn5 affect the level of RPA acetylation in the cell. **D)** HR reporter assay in human U2OS cells containing an inducible I-SceI endonuclease and an integrated DR-GFP reporter. After siRNA transfection, I-SceI DSB induction and repair, cells were subject to fluorescence-activated cell sorting (FACS) analysis for GFP expression (reflecting the level of repair by HR). Results represent the percentage of GFP-positive cells normalized on control siRNA. Double knock-down of Tip60 and hGcn5 shows stronger defect in repair by HR. PALB2 knock-down is shown as positive control. Error bars represent standard error from three biological replicates. **E)** Model depicting the roles of NuA4 and SAGA HAT complexes for the repair of DNA breaks by homologous recombination. It shows the recruitment to the break by MRX, acetylation of chromatin to allow resection with recruitment of remodelers, followed by acetylation of RPA on resected ssDNA and acetylation of chromatin at the homologous donor sequence during strand invasion. (Statistical analyses were performed by one-tailed t-test: * $p<0.05$, ** $p<0.01$, *** $p<0.001$; and multiple comparisons significance for B and D was also confirmed by one-way ANOVA followed by Tukey test).

SAGA complex is recruited to the DNA break in a Xrs2-dependent manner, similar to NuA4. Strikingly, we show that the combined action of the two HAT complexes is essential for DNA end resection to occur after appearance of the DSB, and therefore HR (**Figs 4 and 5**). Several studies have highlighted the cooperation between NuA4 and SAGA in gene transcription through acetylation of chromatin [35–37]. It has been shown that NuA4- and SAGA-dependent local acetylation of chromatin stimulates the association of ATP-dependent chromatin remodelers SWI/SNF and RSC through an interaction of their bromodomains with acetylated

histones both at the promoter and during transcription elongation [47–49]. Since SWI/SNF and RSC complexes are also involved in the repair of DSBs [50] and have been recently shown to be essential for nucleosome eviction and DNA end resection [51], it is not surprising that a similar mechanism can occur following NuA4- and SAGA-dependent acetylation of nucleosomes surrounding the damage site, and that loss of both complexes has a most detrimental effect. Since it was proposed that SWI/SNF, RSC and Ino80 function at different steps of the HR process [30,52–54], it will be interesting to investigate whether the HAT complexes are implicated. Moreover, since transcription at DSBs has been recently implicated in the repair process [55–57], it will be also important to determine if NuA4/SAGA-dependent acetylation can drive transcription near DNA breaks. Based on current and previous results, a model for NuA4 and SAGA function in HR-mediated repair of DSBs is depicted in **Fig 5E**. It highlights their essential role upstream of chromatin remodelers for DNA end resection and implicates NuA4 at the later step of strand invasion during HR.

Important questions remain about NuA4 and SAGA involvement/mechanisms in different DNA repair pathways. NuA4-dependent acetylation of RPA has been suggested to destabilize RPA from ssDNA [5]. Our results indicate that NuA4 and SAGA again cooperate in that process, which in turn could favour RPA replacement by the Rad51 recombinase. This is difficult to judge in our system since NuA4 and SAGA are required for resection and therefore RPA loading on the ssDNA. Since RPA antagonizes micro-homology mediated end-joining (MMEJ) [58], it is also possible that NuA4 and SAGA participate in MMEJ repair by regulating RPA dynamics. A study in mammalian cells reported that p400, a subunit of the TIP60/NuA4 complex, regulates a similar alternative end-joining pathway [59], supporting another possible conserved function in repair pathways.

Moreover, other DNA damage response factors are substrates of HATs, such as Rad50 and Sae2 [40,60]. NHEJ factors are also known to be acetylated, including Ku and Nej1 [61,62]. It will be interesting to determine how NuA4 and SAGA may directly regulate these factors and differentially affect HR and NHEJ pathways. The literature and our data point to an essential role of histone acetyltransferase activities in stimulating DSB repair by the HR pathway, while histone deacetylases have been reported as major facilitators of NHEJ [63]. This highlights an important function of chromatin structure in regulating resection which is the key step in repair pathway selection. Since NuA4 and SAGA are highly conserved during evolution and involved in several human diseases [10,64], it will be of interest to dissect the actions of each HAT, investigate the potential functional redundancy in these acetylation events and elucidate the multifaceted HAT collaboration from yeast to human. The development of HAT inhibitors with greater specificity will allow efficient synthetic lethal screens to target specific cancer types in order to cripple their repair potential during radio- and chemotherapy [65]. Interestingly, Tip60/KAT5 has recently been identified through a CRISPR screen as a key regulator of PARP inhibitor response in BRCA2-deficient cells [66].

## Methods

### Yeast strains

Yeast strains used in this study are listed in **S1 Table** and were constructed based on conventional PCR-based transformation protocol. Detailed construction information is available upon request. For N-terminal tagging of proteins, 500bp upstream and 500bp downstream sequence from start codon was inserted in pFA6a-kanMX6 plasmid [67] after the kanMX6 cassette using Gibson assembly cloning (NEB, E2611). FRB tag [24] was then inserted after the start codon by using Gibson assembly cloning and used as template for PCR. Cells were transformed with PCR product using forward primer (40bp before start codon

+ GAATTCGAGCTCGTTTAAAC) and reverse primer (~20bp reverse complement sequence before +500bp) and selected for G418 resistance. The region corresponding to the N-terminal region of the protein was sequence verified. The yeast strain used for transformation was previously mutated (*tor1-1* mutation and *FPR1* deletion) to make the cells resistant to rapamycin, while Rpl13a was tagged with FKBP12, as described [24]. BrdU-inc strains were constructed using p405–BrdU–Inc as reported [68]. For *MAT*-inc strains used in Figs 5A, 5B and S2C and S2D, *MATα* sequence was mutated at HO cut site and then inserted to the *HIS3* locus using *URA3* selection. Yeast growth/sensitivities by serial dilution spot assays and flow cytometry analysis of cell cycle were performed as before [5].

## Chromatin immunoprecipitations (ChIP)

ChIP-qPCR was performed essentially as reported previously [5] except for the following modifications. For experiments using *esa1-L254*P mutant, since cells tend to accumulate in G2/M [11], cells were synchronized in G2/M before induction of HO with galactose (to avoid potentially differed cell cycle population between WT and mutant). More specifically, cells were grown overnight in YPR at 23°C until OD600 around 0.5. Nocodazole was then added (20μg/ml final in 1% DMSO) and cell culture was shifted to 32°C for 3hrs before addition of 2% galactose for another 3hrs at 32°C. Cells were cultured at 30°C for all the other ChIP-qPCR experiments performed. For experiments with anchor-away system, cells were grown overnight in YPR until OD600 around 0.5, and rapamycin was added to a final concentration of 1ug/ml for 20min followed by 2% final galactose induction for 3hrs. For experiment with BrdU, cells were grown overnight in YPR until OD600 around 0.5 and then BrdU was added (400μg/ml final) and grown for 6hrs. HO cleavage efficiency was calculated with the ratio from qPCR using primers across the DSB site over an uncut control locus (interV, long intergenic region on chromosome V). Lists of primers standardised on a LightCycler qPCR apparatus are available upon request. Antibodies used were against penta-H4 acetylation (Upstate, 06–946), H4 (Abcam, Ab7311), H2B (Abcam, Ab1790), Myc (9E10), Rad51 (SCBT, sc-33626), BrdU (GE, RPN202), Eaf1 [12], Swi2/Snf6, Sth1/Rsc6 and Rvb1 (kindly provided by Joe Reese, Brad Cairns and Yasutaka Makino, respectively). ChIP-qPCR data are presented as % of IP/input (or ratio of IP/input when correcting for nucleosome occupancy, i.e. H4ac/H3, H2B or H4) and are from at least two independent yeast cultures in each experiment. Anti-Rfa1 ChIP-coupled to mass spectrometry was described in [5].

## Plasmid-based NHEJ assay

Plasmid-based NHEJ assay was performed essentially as described previously [69]. After transformed with cut or uncut plasmid, cells were plated on URA- plates containing 1μg/ml rapamycin and let grow around 3 days before the colony numbers were quantified. *LEU2* ORF was completely removed from HHY168 genome to prevent potential repair through HR.

## Acetyl-lysine IP

Immunoprecipitation with acetyl-lysine antibody was performed essentially as described previously [5]. Cells were grown in YPD until OD600 around 0.5 followed by addition of rapamycin (1ug/ml final) and DMSO or MMS (0.05% final) for 3hrs. Antibody used were acetyl-lysine (ImmuneChem, ICP0380), RPA (Agrisera, AS 07–214) and Pgk1 (Abcam, ab113687).

## DR-GFP HR assay

DR-GFP HR assay was performed with DR-GFP-integrated U2OS cells, essentially as reported previously [20]. Knockdown (KD) was carried out with siRNA transfection using

Lipofectamine RNAimax (ThermoFisher, 13778075) following manufacture's instruction. siTip60 was a single siRNA from the reported Smartpool siRNA pool [20] and validated for KD efficiency (slightly weaker KD than the Smartpool). siGcn5 and siPCAF used were as reported previously [45]. Specific sequences of siRNA are available upon request.

## Key points

- NuA4-dependent acetylation modulates nucleosome dynamics to assist DNA end resection and strand invasion during homologous recombination.

- Combined recruitment and local action of NuA4 and SAGA acetyltransferase complexes is essential for DNA end resection.

- Cooperation between NuA4 and SAGA acetyltransferases is required for homologous recombination in eukaryotes.

## Supporting information

**S1 Fig. Using NuA4 mutants in the Gal-HO system to study DSB repair.** Related to Fig 1. **A)** Percentage of HO cutting efficiency in WT and *esa1-L254P* mutant after 2hr 37˚C treatment followed by 3hr galactose induction at 37˚C. **B)** Western blot (WB) of whole cell extract (WCE) from WT and *esa1-L254P* cells treated with 23˚C or 32˚C for 3hr. Antibodies used are penta-H4 acetylation (Upstate, 06–946) and H4 (Abcam, Ab7311). **C)** Percentage of HO cutting efficiency in WT and *esa1-L254P* cells after 3hr galactose induction at 32˚C (based on qPCR across the cut site normalised to control locus). **D)** ChIP-qPCR of H4ac signal at the *RPS11B* control locus in WT and *esa1 ts* strains during galactose induction of HO nuclease, showing no increase (ratio of H4ac/total H3%IP/input, error bars are range of biological replicates). **E)** Western blot (WB) of H4 acetylation and H3 with whole cell extract (WCE) from indicated strains grown in YPD without rapamycin treatment, showing the non-specific effect of Esa1 tagging at its C-terminus. **F)** HO cutting efficiency as above in No-FRB and FRB-Esa1 cells after 30min of galactose induction in presence of rapamycin (depleted conditions, no change is seen after 3hrs either). **G)** ChIP-qPCR of H4ac signal as above at the *RPS11B* control locus in No-FRB and FRB-Esa1 strains during galactose induction of HO nuclease. **H)** Relative H4ac/total H4 level by ChIP-qPCR (ratio of % IP/input) in No-FRB cells after 3hr galactose induction in presence of rapamycin. Values at the different locations near the DSB at time 0 was normalized to 1, showing the increase of H4 acetylation upon HO cut. **I)** Relative H4ac/total H4 level by ChIP-qPCR (ratio of % IP/input) in No-FRB and FRB-Esa1 cells after 3hr galactose induction in presence of rapamycin. Values at the different locations near the DSB in No-FRB cells were normalized to 1, showing the loss of H4 acetylation induction in FRB-Esa1 cells. **J)** Relative total H4 level (nucleosome occupancy) by ChIP-qPCR in No-FRB and FRB-Esa1 cells after 0 and 3hr galactose induction in presence of rapamycin. t = 0 values of respective strains were normalized to 1 to show the loss of nucleosomes upon DSB induction, which is less important after depletion of Esa1. Error bars represent range from biological duplicates. **K-L)** BrdU assay validation for measurement of end resection. **K)** Dot blot of

BrdU-incorporation strains (BrdU-inc +) or non-modified strains (BrdU-inc -) grown in YPD without (BrdU -) or with (BrdU +) 400μg/ml BrdU for 4hrs. Genomic DNA was spotted on Amersham Hybond-N+ membrane and blotted with BrdU antibody (GE, RPN202). **L)** ChIP assay of BrdU (%IP/input) in No-FRB strain after 0 (No damage) or 3hr (+damage) galactose induction, showing clear detection of ssDNA generated by resection within 3kb of the HO break with background signal at 10kb and control locus and in cells not incubated with BrdU (no BrdU).
(TIF)

**S2 Fig. NuA4 *esa1* mutant or depletion through Eaf1 does not affect NHEJ and HR in reporter assays measuring growth on plates.** Related to Figs 2 and 3. **A)** Plasmid-based NHEJ assay using indicated strains transformed with cut or uncut plasmids, plated on 1μg/ml rapamycin-containing SCSM-URA solid medium and grown at 30˚C. Error bars represent standard errors from biological triplicates. **B)** 10-fold serial dilutions of indicated strains spotted on solid medium without or with indicated amounts of MMS and grown at 32˚C, showing synthetic sickness/interaction between *esa1 ts* mutant and *fun30* deletion. **C-D)** 10-fold serial dilutions of indicated strains were spotted on YPD or YP containing 2% galactose to induce HO break and recombination with the mat-inc donor locus (measured by survival, see system schematic in Fig 5A). The *esa1 ts* mutant shows no defect in HR at 32˚C (*rad52* deletion is used as positive control) (C). The Eaf1-FRB strain does not show HR defect either in presence of 1μg/ml rapamycin grown at 30˚C (D).
(TIF)

**S3 Fig. Effects of anchor-away depletion of Esa1, Gcn5 and Esa1/Gcn5 together.** Related to Fig 4. **A)** Western blot analysis of bulk histone acetylation marks in No-FRB, FRB-Esa1, Gcn5-FRB and FRB-Esa1/Gcn5-FRB cells after 0, 60, 120, 180 and 240min in Galactose to induce the HO break as well as in the presence of rapamycin (added 20min before time 0 of galactose). A no rapamycin control before galactose is also shown. Esa1 depletion shows rapid loss of H4ac (penta) while Gcn5 depletion shows a decrease of H3ac, most notably H3K27ac. **B)** Percentage of HO cutting efficiency in No-FRB, FRB-Esa1, Gcn5-FRB and FRB-Esa1/Gcn5-FRB strains after 30min of induction in galactose. Error bars represent standard error from biological triplicates. Error bars represent standard errors from biological triplicates. **C-D)** Cell cycle analysis of cells from the same indicated strains by cell cytometry after fixing and staining with PI, before (C) and after 3hr induction of HO (D) in presence of rapamycin, showing no major changes in cell cycle profiles between strains and conditions. Error bars represent range from biological duplicates. **E)** ChIP-qPCR of H4ac signal (ratio of H4ac/total H4%IP/input) next to the HO break in No-FRB, FRB-Esa1 and FRB-Esa1/Gcn5-FRB strains after 3hr of galactose induction of HO in presence of rapamycin, showing the expected loss of H4ac. **F)** ChIP-qPCR of relative H4 level (nucleosome occupancy) in the indicated strains after 3hr galactose induction in presence of rapamycin. Values (% IP/input) near the HO break were compared to the No-FRB samples set to 1, showing an increase in Esa1/Gcn5-depleted cells. Arp4-FRB is shown as control as it is a shared subunit not only of NuA4 but also INO80/SWR1 chromatin remodeling complexes. Error bars represent range from biological duplicates.
(TIF)

**S4 Fig. The ATM/ATR-related Tra1 subunit shared by NuA4 and SAGA affects DSB repair but does not recapitulate combined Esa1/Gcn5 function.** Related to Figs 4 and 5. **A)** 10-fold serial dilutions of indicated FRB-tagged strains were spotted on solid medium without or with 1μg/ml rapamycin supplemented without or with 0.03% MMS and grown at 30˚C. Depletion

of the Tra1 subunit shared by NuA4 and SAGA leads to much decreased viability, while not as much as Esa1/Rpo21, but also clear sensitivity to DNA damage. **B)** Percentage of HO cutting efficiency in No-FRB and FRB-Tra1 strains after 30min of induction in galactose. Error bars represent standard error from biological triplicates. **C-D)** ChIP-qPCR assay of Rfa1-13myc (C) and Rad51 (D) (% of IP/input at different locations around the HO DSB) in No-FRB and FRB-Tra1 strains after 3hr of galactose induction. Error bars represent range from biological duplicates.
(TIF)

**S5 Fig. Knockdown of PCAF accumulates cells in S/G2.** Related to Fig 5D. **A)** DR-GFP as in Fig 5D with indicated siRNAs. **B)** Cell cycle analysis of cells shown in (A) and Fig 5D by cell cytometry after fixing and staining cells with PI. Error bars represent range from biological duplicates. Note that siPCAF cells show accumulation in S/G2, potentially accounting for elevated HR repair efficiency observed in these cells.
(TIF)

**S1 Table. Yeast strains used in this study.**
(DOCX)

## Acknowledgments

We thank Jean-Phillippe Côté for technical assistance. We are very grateful to Joe Reese for providing Swi2 and Snf6 antibodies, Bradley Cairns for Sth1 and Rsc6 antibodies, and François Robert, Lorraine Symington and Susan Gasser for sharing yeast strains.

## Author Contributions

**Conceptualization:** Xue Cheng, Jacques Côté.

**Data curation:** Xue Cheng, Valérie Côté.

**Formal analysis:** Xue Cheng, Valérie Côté, Jacques Côté.

**Funding acquisition:** Jacques Côté.

**Investigation:** Xue Cheng, Valérie Côté.

**Methodology:** Xue Cheng.

**Project administration:** Jacques Côté.

**Supervision:** Jacques Côté.

**Validation:** Jacques Côté.

**Writing – original draft:** Xue Cheng, Jacques Côté.

**Writing – review & editing:** Jacques Côté.

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
