## [Decision Letter · Decision Letter 0]

23 Mar 2021

Dear Jacques,

Thank you very much for submitting your Research Article entitled 'NuA4 and SAGA acetyltransferase complexes cooperate for repair of DNA breaks by homologous recombination' to PLOS Genetics. The manuscript was fully evaluated at the editorial level and by three independent peer reviewers. The reviewers were enthusiastic about the results and their potential impact, but identified some concerns that we ask you address in a revised manuscript.  In particular, statistical tests need to be incorporated throughout and protein depletion needs to documented.  In addition, the black and gray/blue lines in Figure 1A-B were difficult to distinguish and many of the panels were missing a Y-axis line. 

We ask you to modify the manuscript according to the review recommendations and your revisions should address the specific points made by each reviewer.

[LINK]

Yours sincerely,

Sue Jinks-Robertson, Ph.D.

Associate Editor

PLOS Genetics

John Greally

Section Editor: Epigenetics

PLOS Genetics

Reviewer's Responses to Questions

**Comments to the Authors:**

Reviewer #1: The article by Cheng et al describes new insights to the role of NuA4, in cooperation with SAGA in DSB repair. Interestingly these effects are both at the broken end, regulating resection and Rad51 recruitment, and at the donor sequence where strand invasion occurs. The authors generate and validate some nice tools for analysis of NuA4 function that complements their work with ts-alleles (e.g. FRB-Esa1). Most importantly they define a exciting new cooperative mechanism between SAGA and NuA4 in regulating end resection. The results with the doubly-depleted cells are among the strongest and most compelling in the paper. They also nicely link these changes in acetylation to downstream effects on chromatin remodeler recruitment and to non-histone substrates (RPA). Overall, I really liked this study and think it contributes important new information. Below I highlight a few potential improvements.

1. While the trends are clear enough, the authors need to apply some kind of statistical comparison to data in Figure 1 to highlight differences. This is true for all quantitative tests in the paper – some statistical analysis is needed, but I did not see any. Given the subtle effects, there may not be significant differences between the single mutants and WT, but I think the strength of the results in Figure 4 and 5 still make the most important contribution.

2. I did not follow the logic on line 132 that decreased Rad51 recruitment supports a role for NuA4 after resection. Wouldn’t reduced resection lead to shorter RPA filaments and therefore less Rad51 signal in bulk ChIP experiments? I think this could be reworded.

3. There are no molecular weight markers on any of the western blots in the paper. In Figure 5C it is not even clear which RPA subunit is being blotted? Which protein is modified? In addition, can the authors say more about acetylation of RPA? I wasn’t aware of this modification and I wondered if it cooperates with RPA sumoylation or phosphorylation that can be modified by DNA damage.

4. Histone acetylation promotes transcription in other contexts. Transcription has recently been ascribed various important roles in DSB repair. For example, RNA:DNA hybrids are implicated in end resection, RNAPIII and the MRN/X complex have both been implicated in regulating transcription at DSB ends. The authors should say more about these connections to their work and consider whether the acetylation effects are important for driving transcription near breaks. See PMIDs: 33406426; 33626331 and other slightly older work on transcription-associated HR (TA-HRR).

Reviewer #2: Cheng, Cote and Cote investigate the roles of two KAT complexes, NuA4 and SAGA, in DSB in yeast. NuA4 has been demonstrated to be recruited to DSBs and to aid in recruitment of both repair complexes and chromatin remodeling complexes. Here the authors aim to further delineate these functions using a conditional mutation and a rapid depletion approach. They also compare the effects of Esa1 (NuA4) and Gcn5 (SAGA) alone and together. Although these approaches are useful and may give new insights, several issues limit the impact of the paper as currently presented.

1. Although many of assays (e.g. HO-induced DSBs) are well-established, they still should be described well enough that newcomers can understand what the assay is and why it is used. The descriptions of all assays in Figure 1 need to be more clearly explained in the text, to set the stage for that figure and subsequent figures in the paper.

2. Use of the rapid degrader system is a good approach. However, I could not find a blot showing the timing and level of knock down for FRB-Esa1 or FRB-Gcn5. These should be added to the blots shown in Suppl 3B. This is especially important for Gcn5, since there seems to be little or no effect on H3Ac in the FRB-Gcn5 strain.

3. Th effects of the esa1 ts mutant on resection are described (line 116) as “small but significant”. How was significance established? Also, in Fig 1E, the red line (mutant) is higher than the black line (WT)—doesn’t that indicate more, not less, resection in the mutant?

4. The “significance” question is relevant to all assays and time points, in all graphs,

in Fig 1.

5. A brief introduction to the resection factors used in the genetic assays in Fig 2 would also help the non-expert reader. Just a line or two about what the factors are what steps they are known to execute in resection.

6. The effects of the double esa1 gcn5 mutants are interesting, but these KATs affect many processes, including transcription and cell cycle progression, in addition to DNA repair. The authors need to demonstrate that the effects are not secondary to the cells being really “sick”—i.e. show some process that is NOT affected by the double mutant. Or show that the effects are not coupled to or downstream of transcriptional defects. They should also directly address exactly how sick the double mutants are.

7. Minor point—the authors in the beginning point out that NuA4 has been implicated in multiple types of DNA repair. They should also mention other types of repair that involve Gcn5, for example NER in mammalian cells, together with E2F.

Reviewer #3: The authors use a conditional mutant and rapid depletion approach along with an integrated double strand-break induction system to show, in yeast, that SAGA and NuA4 cooperate to acetylate chromatin around sites of DNA damage and at donor sites. Cooperation between these chromatin modifying complexes is important for DNA end resection to occur, recruitment of chromatin remodeling complexes, acetylation of DNA repair factors, and homologous recombination.

This work was a pleasure to read and the discussion clearly points where the next papers should go. Comments on minor issues are as follows:

Minor points:

1. Make the Y-axis for Figures 1 A and B the same, so they can be easily compared.

2. “Interestingly, both systems indicate that the domain of NuA4-dependent increased H4 acetylation upon DSB formation is larger than the extent of NuA4 spreading in these conditions (3-5kb, see [5]), suggesting longer-range contact in the tri-dimensional space.”

a. Could the authors explain this more? How would there be acetylation without NuA4 spreading? There is a distinction being made between spreading and contact, which implies NuA4 is responsible for acetylation in regions where it can not be shown to interact.

3. “Accordingly, in the ts mutant, we detect a small but significant decrease of DNA end resection near the break site as well as decreased single-strand DNA (ssDNA) binding RPA and Rad51 recombinase recruitment (Fig. 1E-G).”

a. Figure 1E shows a small increase in DNA end resection near the break site. Are the labels wrong? The data in 1H supports the text and contradicts 1E.

b. Figure 1F is labeled Myc-Rfa1, and so does the figure legend, but the manuscript text refers to RPA

4. It is interesting that the Rfa data in Figure 1 shows decreased ssDNA near and far from the break point, but the Rad51 data shows it more near the break site. Please comment on this.

5. “(Fig. 3F). These results provide the first evidence that links NuA4 to D-loop formation” The term “links” suggests some need or influence of NuA4, but at this point in the paper, you have shown that NuA4 and H4ac are present at the site of D-loop formation.

6. Quantify 4E and 5C

**Have all data underlying the figures and results presented in the manuscript been provided?**

Reviewer #1: Yes

Reviewer #2: Yes

Reviewer #3: Yes

PLOS authors have the option to publish the peer review history of their article (what does this mean?). If published, this will include your full peer review and any attached files.

Reviewer #1: No

Reviewer #2: No

Reviewer #3: **Yes: **Ryan D Mohan

---

## [Decision Letter · Decision Letter 1]

10 Jun 2021

Dear Jacques,

Thank you very much for submitting your revised Research Article entitled 'NuA4 and SAGA acetyltransferase complexes cooperate for repair of DNA breaks by homologous recombination' to PLOS Genetics. The reviewers appreciated the care taken in addressing their previous comments and uniformly recommended acceptance. That being said, reviewer 1 asked that you address two statistic-related issues before final acceptance. Once this is done, I will be happy to accept the manuscript with no further review. Please just let me know your responses to the review comments and provide a description of any changes made to the manuscript.

Also, consider uploading a Striking Image with a corresponding caption to accompany your manuscript if one is available (either a new image or an existing one from within your manuscript). If this image is judged to be suitable, it may be featured on our website. Images should ideally be high resolution, eye-catching, single panel square images. For examples, please browse our archive. If your image is from someone other than yourself, please ensure that the artist has read and agreed to the terms and conditions of the Creative Commons Attribution License. Note: we cannot publish copyrighted images.

[LINK]

Yours sincerely,

Sue Jinks-Robertson, Ph.D.

Associate Editor

PLOS Genetics

John Greally

Section Editor: Epigenetics

PLOS Genetics

Reviewer's Responses to Questions

**Comments to the Authors:**

Reviewer #1: The revision is clearer and much improved. Thank you for adding statistics. On that point, I would just encourage the authors to check two things.

1. When multiple comparisons are being made an ANOVA followed by post-hoc test should be used to correct for testing multiple hypotheses. For example in Figure 3 when two bars are compared to the first one.

2. When data trends are the same but do not reach statistical significance (e.g. Figure 3F has no stars), then I think the authors should change their language a bit more. So in 3F the authors say that there is a concomitant increase in H4 acetylation detected at the donor sequence. I agree its a strong trend, but fails to meet significance due to presumably variability. So stating that there was a doubling of the ChIP signal that failed to reach statistical significance would be more accurate.

Reviewer #2: The authors have carefully addressed all previous concerns and suggestions. The paper is much more accessible and will be of interest to a wide audience.

Reviewer #3: Revisions are acceptable. Well done.

**Have all data underlying the figures and results presented in the manuscript been provided?**

Reviewer #1: Yes

Reviewer #2: Yes

Reviewer #3: Yes

PLOS authors have the option to publish the peer review history of their article (what does this mean?). If published, this will include your full peer review and any attached files.

Reviewer #1: No

Reviewer #2: No

Reviewer #3: **Yes: **Ryan D. Mohan

---

## [Editor Report · Decision Letter 2]

21 Jun 2021

Dear Jacques,

We are pleased to inform you that your manuscript entitled "NuA4 and SAGA acetyltransferase complexes cooperate for repair of DNA breaks by homologous recombination" has been editorially accepted for publication in PLOS Genetics. Congratulations!

Yours sincerely,

Sue Jinks-Robertson, Ph.D.

Associate Editor

PLOS Genetics

John Greally

Section Editor: Epigenetics

PLOS Genetics

**Data Deposition**

http://datadryad.org/submit?journalID=pgenetics&manu=PGENETICS-D-21-00284R2

**Press Queries**

---

## [Editor Report · Acceptance letter]

30 Jun 2021

PGENETICS-D-21-00284R2 

NuA4 and SAGA acetyltransferase complexes cooperate for repair of DNA breaks by homologous recombination 

Dear Dr Côté, 

We are pleased to inform you that your manuscript entitled "NuA4 and SAGA acetyltransferase complexes cooperate for repair of DNA breaks by homologous recombination" has been formally accepted for publication in PLOS Genetics! Your manuscript is now with our production department and you will be notified of the publication date in due course.

With kind regards,

Zsofi Zombor

PLOS Genetics

On behalf of:
